# A new taekwondo-specific field test for estimating aerobic power, anaerobic fitness, and agility performance

**Behzad Taati** [1], **Hamid Arazi** [1]*, **Craig A. Bridge** [2], **Emerson Franchini** [3]

**1** Faculty of Sport Sciences, Department of Exercise Physiology, University of Guilan, Rasht, Iran, **2** Sports Performance Research Group, Edge Hill University, Wilson Centre, Ormskirk, United Kingdom, **3** Martial Arts and Combat Sports Research Group, Sport Department, School of Physical Education and Sport, University of São Paulo, São Paulo, Brazil

* hamidarazi@yahoo.com

**Data Availability Statement:** All relevant data are within the manuscript and its Supporting Information files.

**Funding:** The authors received no specific funding for this work.

## Abstract

The present study aimed to propose a new multidimensional taekwondo-specific test to estimate aerobic power, anaerobic fitness, and agility. Out of sixty-five male volunteers, forty-six, forty-eight, and fifty athletes (18–35 years; black- and red-belt level) were included in the final analysis for aerobic, anaerobic, and agility assessments, respectively. Maximum oxygen uptake ($VO_{2max}$, using a graded exercise test on a treadmill), anaerobic power (using the 30-s Wingate anaerobic test, WAnT), and agility performance (using the agility T-Test) were measured via non-specific laboratory and field tests across a two-week period. The taekwondo-specific aerobic-anaerobic-agility (TAAA) test comprised six 20-s intervals of shuttle sprints over a 4-m distance, and the execution of roundhouse kicks alternating the legs at the end of each distance, with 10-s rest intervals between the sets. The multiple linear regression revealed that the difference between heart rate (HR) after and 1 minute after the TAAA test ($p < 0.001$), and body mass index (BMI; $p = 0.006$) were significant to estimate $VO_{2max}$. Likewise, there was a very large ($R = 0.79$) and large ($R = 0.55$) correlation between the average and maximum number of kicks performed in the TAAA test and the WAnT mean and peak power, respectively ($p < 0.001$). Moreover, a linear relationship was found between the T-Test and agility performance acquired in the TAAA test ($R = 0.74$; $p < 0.001$). The TAAA test can be considered a valid simple tool for monitoring $VO_{2max}$, anaerobic fitness, and agility in male taekwondo athletes.

## Introduction

Taekwondo is a modern combat sport that has been included in the Olympic Games since 2000 [1,2]. Taekwondo matches typically consist of three 2-min rounds separated by 1-min rest intervals [3], and during the rounds athletes perform high-intensity actions (e.g., scoring or defensive techniques) interspersed by low-intensity periods (e.g., stepping actions or referee's breaks) [3–5]. Taekwondo competitors intend to overcome an opponent through either obtaining a greater quantity of points for the execution of kicks to the torso and head, and by

**Competing interests:** The authors have declared that no competing interests exist.

punches to the torso, or through achieving a technical knockout [3]. During championship matches, competitors perform brief periods of fighting [attacks] (1-6s), interspersed with longer periods of rest (fighting to non-fighting ratios ranging between 1:2–1:7) [3,5]. The contests elicit high heart rate responses (>90% peak heart rate, $HR_{peak}$) and moderate to high lactate concentrations (7.0–12.2 mmol.l$^{-1}$) [3,6,7]. Therefore, the phosphagen system is likely to supply the energy during high-intensity actions, the glycolytic system supports repeated high-intensity actions, whereas the oxidative system is important to facilitate the recovery process between these actions and successive bouts in championship events [3,6,7]. As such, taekwondo competitors need high anaerobic and aerobic power abilities to effectively manage the metabolic demands during the bouts. It has been concluded that aerobic and anaerobic power are crucial performance determinants of success in taekwondo athletes [8]. In this context, Sadowski et al. (2012) reported that medalist male taekwondo athletes demonstrated higher anaerobic power on the Wingate test than their non-medalist counterparts [9]. Furthermore, there was a tendency for senior and junior male medalists to show a higher aerobic power on the multistage shuttle run test than their non-medalist counterparts [9,10].

Moreover, agility is another important factor for taekwondo athletes to achieve high-performance success [3,11,12]. Agility is defined as a quick whole-body movement including both deceleration and acceleration phases with a change of speed or direction in response to a given stimulus (12). This capability is considered crucial to rapidly perform all-out technical-tactical actions in multidirectional planes while dynamic balance, speed, and precision are maintained (11). Indeed, Marković et al. (2005) reported better agility performance in successful female athletes in comparison to their less successful counterparts [12].

These data highlight the importance of both aerobic and anaerobic systems together with agility, as a prerequisite for success in taekwondo. Effective monitoring of these key physical performance capabilities is therefore necessary to identify strengths and weaknesses in specific physical attributes, to monitor fitness status over time, and verify the effectiveness of training [3]. A range of generic and sports-specific laboratory and field-based tests have been routinely administered to evaluate aerobic and anaerobic power, and agility in taekwondo [13–19]. For instance, running and cycling testing modes have been frequently used to measure or estimate maximum oxygen uptake ($VO_{2max}$), whereas the 30s cycling WAnT has been instrumental in evaluating anaerobic power and capacity in taekwondo [3]. Similarly, a range of general agility tests, such as the T-test, 50m shuttle test and the side step test, have been utilized [3,12]. Whilst a number of these testing modes may be regarded as the 'gold standard' in fitness assessment within the sport, their use has been challenged on the grounds of lacking mechanical specificity to the actions in the sport. In fact, these tests do not assess sport-specific performance but only evaluate general physical fitness qualities [20]. As such, there has been a drive to enhance the specificity and validity of the performance tests in Taekwondo. To this end, Taekwondo-specific tests have recently been suggested to separately evaluate anaerobic power [13,14,16], aerobic fitness [15,17–19], and agility [11] with varying degrees of success.

The ability to directly measure specific components of fitness using either established generic exercise testing modes or preferably using taekwondo-specific test modes might be recommended to enhance the accuracy and validity of test data. Strengths and weaknesses of athletes and their performance development over time can be identified by information from these sport-specific tests, especially for taekwondo which is characterized by complex technical/tactical and physical/physiological demands to support kicking and punching actions during the match [20]. Nevertheless, some athletes and coaches may encounter varying constraints that preclude the implementation of this strategy in the 'real world.' This may include access to expensive and specialist equipment to perform direct measurements (i.e. metabolic and blood analysis), specialist/trained personal to administer sophisticated

procedures (i.e. physiologist, phlebotomist), and time constraints associated with administering multiple test batteries. In an attempt to circumvent some of these constraints, a number of research groups have developed submaximal and maximal field-based methods, which are capable of indirectly estimating specific fitness components in taekwondo, such as $VO_{2max}$ [15,19,21]. There have no attempts to develop a single multidimension field-based test capable of estimating multiple important components of fitness in taekwondo or in other sports. This kind of test would be particularly valuable for taekwondo populations with limited access to specialist equipment, expertise, funding and/or time to assist monitoring fitness. Therefore, this study aims to propose a new field test, based on the specific motor skills of taekwondo, to estimate aerobic power, anaerobic fitness, and agility in taekwondo athletes.

## Materials and methods

### Participants

Overall, sixty-five male taekwondo athletes volunteered to participate in the study. However, forty-six, forty-eight, and fifty athletes were included in the final analysis for aerobic, anaerobic, and agility assessments, respectively (Fig 1).

All athletes were recruited from the martial arts clubs of Tehran and Karaj, Iran. They were aware of possible risks and benefits, and signed the written informed consent form before beginning the study. The inclusion criteria included the following: (i) being aged 18 to 35 years; (ii) black- or red-belt level; (iii) regular taekwondo training at least two days per week; and (iiii) having a minimum training experience of three years (Table 1).

The athletes were of different weight categories (Table 2). They participated regularly in regional and province competitions and nine athletes were national competitors (i.e. a member of the national taekwondo team). It also should be mentioned that all athletes were in the general phase of their training program and most of the regular exercises were performed at 60–85% of maximal heart rate ($HR_{max}$). This study was conducted in accordance with the

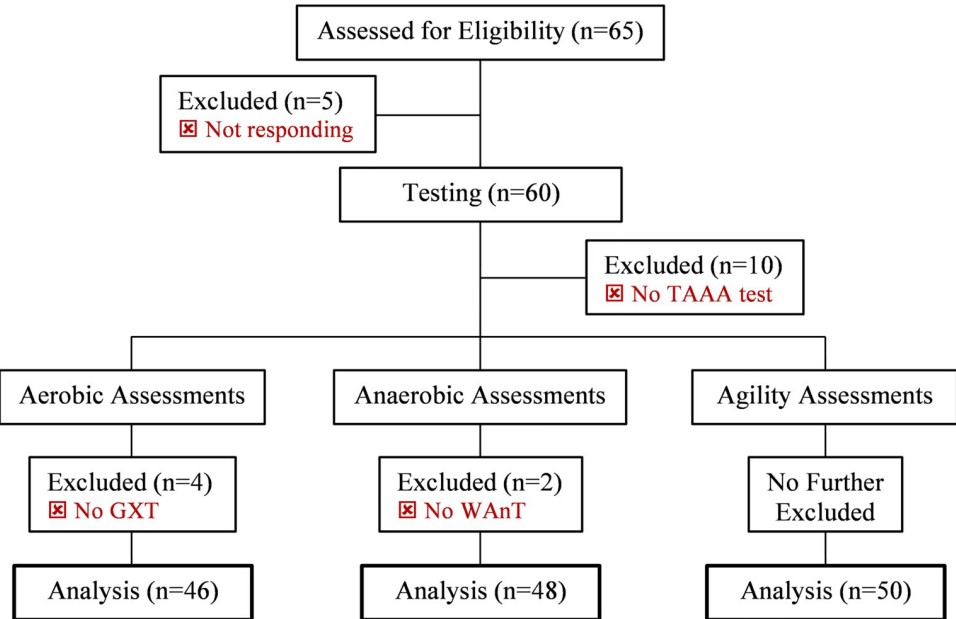

**Fig 1. Flow chart of study progress.** TAAA: Taekwondo-specific aerobic-anaerobic-agility test; GXT: Graded exercise test; WAnT: Wingate anaerobic test.

**Table 1. Demographic characteristics of the athletes participating in different parts of assessments performed in this study*.**

|  | Aerobic assessments (n = 46) | Anaerobic assessments (n = 48) | Agility assessments (n = 50) |
|---|---|---|---|
| Age (years) | 21.3 (5.1) | 20.9 (4.9) | 21.1 (5.0) |
| Height (cm) | 178.6 (5.9) | 179.2 (6.1) | 179.3 (6.0) |
| Body mass (kg) | 70.5 (7.2) | 71.9 (9.7) | 73.5 (12.0) |
| BMI (kg/m$^2$) | 22.12 (2.16) | 22.36 (2.7) | 22.84 (3.51) |
| Body fat (%) | 13.2 (5.0) | 13.0 (4.8) | 13.4 (5.0) |
| Taekwondo experience (years) | 11 (5) | 11 (5) | 10 (5) |

* Data are presented as mean (standard deviation).

Declaration of Helsinki and was approved by the Review Board of Sport Sciences Research Institute (IR.SSRC.REC.1398.122).

## Procedures

Each athlete attended four testing sessions in a randomized manner across a two-week period: VO$_{2max}$ measurement, anaerobic testing session, the taekwondo-specific test session, and agility assessments (i.e. agility performance assessed via the taekwondo-specific test, and agility T-Test). Anthropometric measurements were performed at the beginning of the first testing session in the laboratory. A minimum period of three days was allowed between the sessions and each athlete completed their testing at the same time of day in an indoor facility with a temperature of 24–25˚C. All athletes received some recommendations on how to maintain a balance and routine diet over 24 h prior to each testing day. In addition, they all were instructed to avoid severe physical activity for the previous 24 h.

## Anthropometric measurements

At the beginning of the first testing session, body mass, height and body mass index (BMI) were measured with a scale (BSM370, InBody Co., Ltd., Seoul, South Korea). Skinfolds thickness was also measured using a calibrated caliper (85310, Saehan Tech Co., Ltd., Buchon, South Korea) at three sites including chest, abdomen and thigh on the right side of the body. The measurement of each site was done three times using the sequence mentioned above. Body fat percentage was then calculated using the equations provided by Jackson and Pollock [22], and Siri [23].

**Table 2. The distribution of athletes in different weight and age categories*.**

|  |  | Aerobic assessments (n = 46) | Anaerobic assessments (n = 48) | Agility assessments (n = 50) |
|---|---|---|---|---|
| Weight categories | Under 58 kg | 2 (4.3%) | 2 (4.2%) | 2 (4.0%) |
|  | Under 63 kg | 11 (23.9%) | 10 (20.8%) | 10 (20.0%) |
|  | Under 68 kg | 8 (17.4%) | 8 (16.7%) | 8 (16.0%) |
|  | Under 74 kg | 11 (23.9%) | 13 (27.1%) | 13 (26.0%) |
|  | Under 80 kg | 6 (13%) | 6 (12.5%) | 6 (12.0%) |
|  | Under 87 kg | 5 (10.9%) | 5 (10.4%) | 5 (10.0%) |
|  | Over 87 kg | 3 (6.5%) | 4 (8.3%) | 6 (12.0%) |
| Age categories | 18–25 years | 37 (80.4%) | 40 (83.3%) | 41 (82.0%) |
|  | 26–30 years | 5 (10.9%) | 5 (10.4%) | 5 (10.0%) |
|  | 31–35 years | 4 (8.7%) | 3 (6.3%) | 4 (8.0%) |

* Data are presented as number (percent).

## Graded exercise test

Based on a predetermined time schedule, each athlete attended the laboratory between 10 am to 2 pm in a fasted state ~ 4 hours prior to commencing testing [24]. In order to standardize the later meal, all athletes were instructed to eat a light breakfast (i.e. 2–3 units of bread, 1 unit of white cheese, and a cup of black tea containing white sugar) until 6:30 am. They were also asked to refrain from consuming any other meals after the breakfast until the end of the test.

Following anthropometric measurements, the athletes completed a graded exercise test (GXT) on a treadmill (T150, COSMED, Italy). The GXT was adopted from previously validated protocol [25]. As described before [2], the test started with a 3-min walking stage at an initial velocity of 5.0 $km.h^{-1}$ and a gradient of 1%. During the second 3 minutes, the speed increased to 7.5 $km.h^{-1}$ while the gradient remained constant. From this point, the speed was increased by 1 $km.h^{-1}$ every 3 minutes until a respiratory exchange ratio (RER) of 1 was reached. The speed then remained constant, and the gradient was increased by 1% every 1 min until the participant reached voluntary exhaustion. A maximum test was defined based on satisfying two of the three following criteria: (i) leveling off in $VO_2$ with further increases in workloads ($<2$ $mL.kg^{-1}$ body mass); (ii) heart rate (HR) within 10 bpm of age-predicted maximum (220 –age), or (iii) RER exceeded 1.10 [2,25]. A warm-up including walking on the treadmill with no gradient for five minutes and lower limb stretching was performed before initiating the GXT protocol.

Breath-to-breath expired air was continuously measured throughout the GXT period, using a calibrated metabolic analyzer (*ZAN-600*, ZAN Messgeräte, Oberthulba, Germany), with 10 s average data recorded upon test completion [2]. Continuous changes in HR were measured by a wireless HR monitor (Polar Xtrainer Plus, Polar Electro Ltd., Kempele, Finland), and the rating of perceived exertion (RPE) was also recorded at the final 30 s of each protocol stage using the Borg 6–20 RPE scale [26].

## Wingate anaerobic test

A standard 30-s WAnT was conducted on a friction belt cycle ergometer (Monark 894 E Peak Bike, Weight Ergometer, Vansbro, Sweden, Software version 2.2) between 10 am and 2 pm according to a predetermined time schedule for each athlete to evaluate the power output. The instructions provided to standardize the later meal were the same as those recommended for the GXT session. Prior to the WAnT, all athletes completed a warm-up period consisting of 5-minutes of cycling on the cycle ergometer and stretching exercises. The WAnT consisted of lower limb cycling with the highest possible number of revolutions per 30 s. The load for each athlete was considered as 7.5% of the body mass [16,27]. The WAnT began from a rolling start against minimal resistance and was performed against the abovementioned constant resistance [27]. All athletes were strongly and consistently encouraged throughout the test to keep the number of revolutions as high as possible. Additionally, they were instructed to maintain a seated posture to avoid the effect of postural changes and to pedal at the maximal effort. Four indices of anaerobic performance expressed as peak power (the highest 5-s output), minimum power (the lowest 5-s output), mean power (average power throughout the test), and anaerobic fatigue or fatigue index (decrease in power output from peak power to minimum power output).

## Agility test

In the present study, the planned test to measure the athletes' agility was the T-Test, which was administered as originally set out by the previous studies [11]. Four cones were arranged in a T shape, so that a cone placed 9.14 m from the starting cone and 2 further cones placed 4.57 m

on either side of the second cone. Athletes started the test with both feet behind the start/finish line. They were asked to touch the base of each cone during the test. After commencement, they had to: (a) sprint forwards to the middle cone and touch with the right hand, (b) shuffle toward the left cone and touch with the left hand, (c) shuffle toward the right cone and touch with the right hand, (d) shuffle back left to the middle cone and touch with the left hand, and (e) back pedaling to the finish line. Failing to touch a designated cone, failing to face forwards at all times, and crossing the legs while shuffling were the criteria for an unsuccessful trial. Each athlete performed 2 trials with a 3-min recovery between them and the shortest record was included in the analysis. A standard stopwatch was used to record time measurement in seconds. Before the study, the researcher had practiced many times to become fully familiar with the device and to reduce the possible error of measurement.

## Taekwondo-specific aerobic-anaerobic-agility test

Taekwondo-specific aerobic-anaerobic-agility (TAAA) test was structured in consideration of the activity typically performed in competition and training, but also in a manner to permit evaluation of the three key fitness components of interest (i.e. agility, aerobic power, and anaerobic fitness). The TAAA test involved six 20-s intervals of shuttle sprints throughout a 4-m distance and performing roundhouse kicks (i.e., *bandal chagi* in taekwondo terminology) alternating the legs at the end of that distance. The main reason to choose the roundhouse kick was that this technique is the most used kick in taekwondo competition and training [6,17]. Based on the real competition time where each round is 2 min and there is a 1-min rest period between the rounds [28], a 10-s rest period was considered between the work intervals to mimic the real time frame in official matches. Likewise, the mentioned distance was also chosen according to the 8 × 8 competition area, so that each shuttle of the TAAA test is equal to 8 m.

The test setup is shown in Fig 2. At first, all athletes performed a warm-up light running followed by stretching exercises and free taekwondo displacements and kicks. Next, each athlete

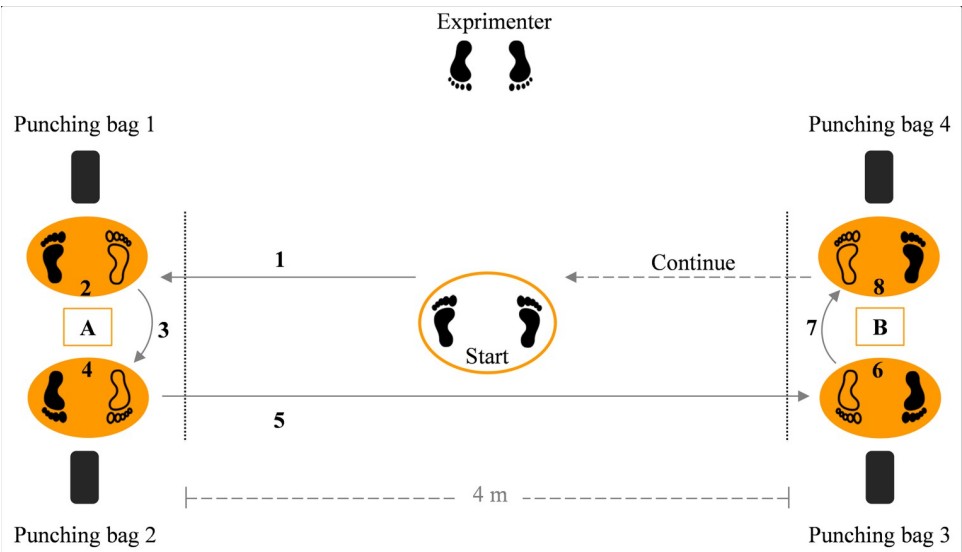

**Fig 2. Taekwondo-specific aerobic-anaerobic-agility (TAAA) test.** The test involves six 20-s (a total of 2 minutes) intervals of shuttle sprints between point A and B and performing roundhouse kicks at these points, with a 10-s rest period between sets. 1) sprint forwards to point A; 2 and 6) a right-foot kick; 3 and 7) a right-side 180-degree rotation; 4 and 8) a left-foot kick; 5) sprint forwards to point B.

started the TAAA test from a ready position with both heels on the ground and one shoulder width between the feet (i.e., *naranhi seogi*) at the point "start", in front of the researcher. After command, the athlete had to: (a) sprint forwards to point A and perform a right-foot kick from a fighting stance, a right-side 180-degree rotation and a left-foot kick from a fighting stance, (b) sprint forwards to point B and perform the same procedure as those at point A. This procedure was continuous between points A and B for the entire 20 s, and the rest periods between the intervals are at the point "start".

In order to estimate anaerobic fitness using the results of TAAA test, the number of 'correct' kicks was recorded during the test as follows:

Maximum kicks: maximum number of kicks in a 20-s interval

Minimum kicks: minimum number of kicks in a 20-s interval

Average kicks: total number of kicks at the end of the test divided by 6

Furthermore, kick fatigue index (KFI) was calculated to determine the amount of the decline in kicking performance during the TAAA test expressed as a percentage of total kicks. The KFI can be calculated using Eq 1.

$$\textbf{Kick Fatigue Index } (\%) = \frac{Maximum\ Kicks - Minimum\ Kicks}{Total\ Kicks} \times 100 \qquad (1)$$

Each kick was deemed correct when the following criteria were achieved: (a) performing behind the lines defining the 4-m distance on both sides, (b) kicking with the instep of the foot at a height between the navel and nipples, (c) kicking with a typical scoring impact. Prior to the test, each athlete performed several kicks with a typical scoring impact to establish a reference kicking impact for subsequent visual inspection by researcher.

The wireless Polar HR monitor (Polar Xtrainer Plus, Polar Electro Ltd., Kempele, Finland) was used to monitor the HR during the TAAA test. The HR after ($H_0$) and one min after ($H_1$) the test was recorded and used as the difference between them ($HR_{diff}$) to estimate $VO_{2max}$ according to the TAAA test.

## Agility performance of the TAAA test

In addition to aerobic and anaerobic estimations, the time needed to perform six correct kicks in the TAAA test was recorded and represented agility performance for each athlete. The shortest time from two attempts with a 3-min rest between them was considered for the analysis. We selected six kicks because the time needed to complete this number of kicks was closer to the records of the planned agility test (i.e. T-Test) during a pilot measurement. This procedure was performed at the agility testing session with a 20- to 30-min rest period between the agility TAAA test and agility T-Test.

## Statistical analysis

All data analyses were performed using SPSS software (v. 18®, SPSS, Inc., Chicago, IL, USA) for Windows and statistical significance was set at $p < 0.05$. At first, normal distribution of data was confirmed using the Kolmogorov-Smirnov (KS) test ($p > 0.05$). Thus, central tendency and spread of the data were reported as mean ± standard deviation (SD).

The multiple regression models were developed to demonstrate the best equation for estimating $VO_{2max}$ and anaerobic power according to the relationship between the measured dependent and independent variables. Pearson's correlation coefficient (*R*) was obtained and the coefficient of determination ($R^2$) was calculated to identify the percent to which the linear

regression equation fits estimated values. The magnitude of $R$ was considered as trivial ($< 0.1$), small ($0.1 < R < 0.3$), moderate ($0.3 < R < 0.5$), large ($0.5 < R < 0.7$), very large ($0.7 < R < 0.9$), nearly perfect ($R = 0.9$), and perfect ($R = 1$) [29]. The standard error of estimate (SEE) was also considered as another criterion to assess the fitting performance of the model. With regard to anaerobic and agility performance, the following categories were adopted using percentile values: excellent ($\geq 95$th percentile); good ($\geq 75$th percentile up to 94th percentile); regular ($\geq 25$th percentile up to 74th percentile); and poor ($\leq 24$th percentile) [30]. The concordance between the agility tests were also assessed by Bland-Altman analysis with the 95% confidence interval (CI). One-sample t test was applied to detect any significant difference of the $\bar{d}$ from zero. Moreover, Levene's F test was also used to determine whether the changes in the values obtained from the TAAA test and the laboratory tests were the same.

In addition, the external responsiveness of the TAAA test for each variable was checked by the receiver operator characteristics (ROC) curves and the area under the ROC curve (AUC) of $> 0.7$ was considered as the good discriminant validity of the test [11,31].

## Results

### VO$_{2max}$

VO$_{2max}$ values (mean ± SD) for the TAAA test and GXT were 52.89 ± 6.18 and 52.89 ± 8.24 mL.kg$^{-1}$.min$^{-1}$, respectively. The mean coefficient of variation (CV) for the TAAA test and GXT was 13.1% and 15.6%, respectively. No significant difference was found in the CV (F = 0.57; $p = 0.45$).

There were no significant correlations between the present predictors including HR$_{diff}$ and BMI ($R = -0.21$; $p = 0.073$), HR$_{diff}$ and age ($R = 0.032$; $p = 0.41$), and BMI and age ($R = -0.091$; $p = 0.27$). The normality assumption of residuals was confirmed using a KS test at 5% α-risk level (KS = 0.08, $p = 0.2$). The multiple linear regression showed that the HR$_{diff}$ (t = 8.97; $p < 0.001$) and BMI (t = -2.88; $p = 0.006$) were significant to estimate VO$_{2max}$. However, the other independent variable, age, did not reach statistical significance (t = 1.77; $p = 0.08$). Therefore, the final adjusted model included two independent variables (i.e. HR$_{diff}$ and BMI), which together explained 70% of the overall variability with a SEE of 4.54 mL.kg$^{-1}$.min$^{-1}$. A very large correlation coefficient was found between the observed and estimated VO$_{2max}$ values ($R = 0.84$; $p < 0.001$). Based on the model, the relationship between VO$_{2max}$ and the independent variables is best described by the following mathematical equation:

$$\textbf{VO}_{\textbf{2max}} \, (\textbf{mL.kg}^{-1}.\textbf{min}^{-1}) = 56.316 + 0.742 \, (\text{HR}_{\text{diff}}) - 0.924 \, (\text{BMI}) \tag{2}$$

### Anaerobic power

Classificatory data for the number of kicks performed in the TAAA test is presented in Table 3. There was a large correlation coefficient between peak power and maximum kicks ($R = 0.54$; $R^2 = 29\%$; $p < 0.001$) and a very large correlation coefficient between mean power

**Table 3. Classificatory results for the number of kicks (n = 48) and agility records (n = 50) obtained from the taekwondo-specific aerobic-anaerobic-agility (TAAA) test*.**

|  | Mean (SD) | Poor | Regular | Good | Excellent |
|---|---|---|---|---|---|
| Maximum Kicks (reps) | 13 (1.25) | $\leq 12$ | 13–14 | 15 | 16 |
| Average Kicks (reps) | 11 (1) | $\leq 10.5$ | 10.6–11.7 | 11.8–12.5 | $12.6 \leq$ |
| Minimum Kicks (reps) | 9.4 (1.14) | $\leq 8$ | 9 | 10 | 11 |
| Agility record (s) | 12 (0.83) | $> 13.5$ | 12.8–13.5 | 11.6–12.7 | $11.5 \geq$ |

* SD: Standard deviation; reps: Repetitions; s: Seconds.

and average kicks ($R = 0.79$; $R^2 = 62\%$; $p < 0.001$), after adjusting for body mass. However, the relationship between minimum power and minimum kicks was not statistically significant ($R = 0.23$; $R^2 = 0.05\%$; $p = 0.1$). Likewise, no significant relationship was observed between power and other independent variables including age and BMI ($p > 0.05$).

The regression models were therefore used to describe the relationship between the correlated variables. The KS test was applied to detect the normality assumption of residuals for maximum (KS = 0.23, $p < 0.001$) and average (KS = 0.09, $p = 0.2$) values. Because this assumption was not met for peak power, the regression model for this variable was not accepted. However, a linear regression model was developed to estimate average power values according to the relationship between average power output obtained from WAnT and average kicks in the TAAA test (t = 8.81; $p < 0.001$), in which the equation that best described this relationship was as follows, where there is no y-intercept of the prediction equation (t = 0.19, $p = 0.85$):

$$\textbf{Average Power } (\textbf{W.kg}^{-1}) = 0.648 \text{ (average kicks)} \tag{3}$$

The abovementioned equation could explain 63% of the overall variability between the variables with a SEE of 0.53 W.kg$^{-1}$. The mean and SD of average power for the TAAA test and WAnT were 7.3 ± 0.85 and 7.3 ± 0.68 W.kg$^{-1}$, respectively. The mean CV was 11.7% and 9.3% for the TAAA test and WAnT, respectively. We then performed the ANOVA Levene´s F test to confirm there was no difference in the CV between the tests (F = 2.34; $p = 0.12$).

The results of KFI related to the TAAA test revealed that the mean and SD of decline in kicking performance was 5.58 ± 2.04% (95% CI = 1.58 to 9.58), with a range of 1.43 to 11.11%.

### Agility records

As presented in Fig 3, after adjusting for body mass, a linear relationship was observed between the T-Test (11.07 ± 0.77 s) and agility performance of TAAA test (12.09 ± 0.83 s). The Pearson's R was 0.73 ($p < 0.001$; $R^2 = 53\%$) which classified as "very large". Hence, classificatory table is presented in Table 3.

The Bland-Altman analysis (Fig 4) and one-sample $t$ test showed there was a significant difference from zero for the amount of $\bar{d}$ between the agility records obtained from the T-Test and TAAA test ($\bar{d} = 1.02$ s; 95% CI = -0.12 to 2.16 s; $p < 0.001$). However, no trend towards a bias was observed for data above or below the $\bar{d}$ of the two tests, according to the linear regression model (t = 0.71; $p = 0.47$). The mean CV for the TAAA test and T-Test was 6.9% and 7%, respectively, with no significant difference (Levene´s F = 0.11; $p = 0.73$).

### External responsiveness of TAAA test

The ROC curve analysis was calculated between national- and regional-level taekwondo athletes to identify the sensitivities and specificities of TAAA test for estimating the variables (Fig 5A–5E). The area under the ROC curve of TAAA test was 0.83 (95% CI = 0.7 to 0.96; p = 0.003), 0.74 (95% CI = 0.58 to 0.9; p = 0.03), 0.75 (95% CI = 0.59 to 0.9; p = 0.02), 0.72 (95% CI = 0.54 to 0.89; p = 0.04), and 0.82 (95% CI = 0.69 to 0.95; p = 0.003) for VO$_{2max}$ (8 national- and 38 regional-level athletes), maximum, average, and minimum kicks (9 national- and 39 regional-level athletes), and agility (9 national- and 41 regional-level athletes), respectively. Therefore, the TAAA test was considered to have very good discriminant ability.

### Discussion

The present study has proposed a new taekwondo-specific field test to indirectly determine anaerobic fitness, VO$_{2max}$, and agility using the most common motor patterns in this combat

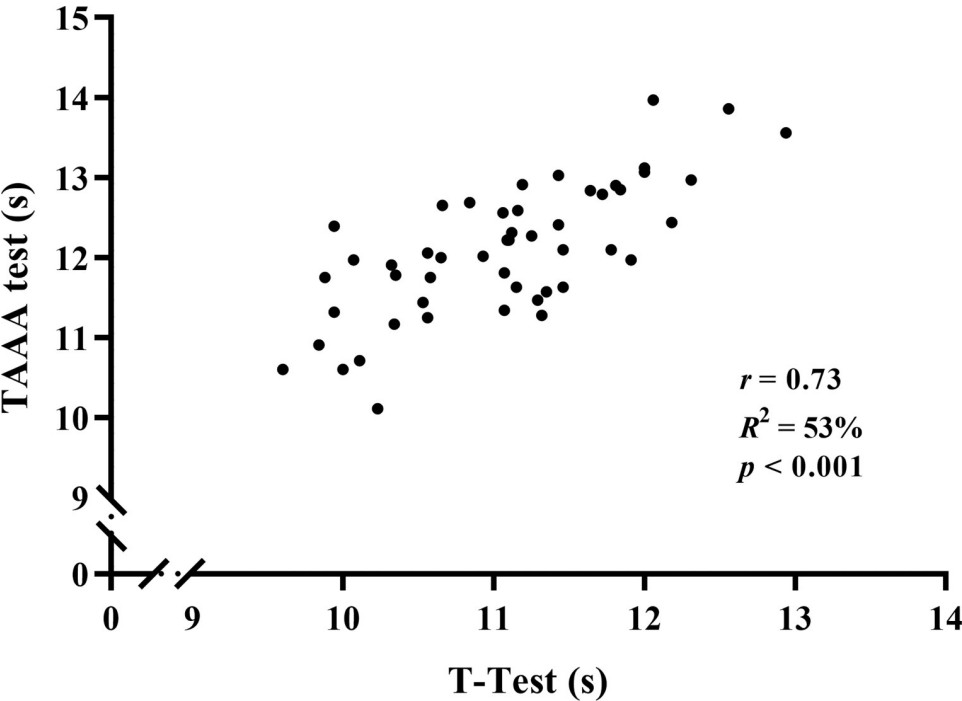

**Fig 3. Scatter plot (n = 50) of the relationship between the agility T-Test and the agility function of the taekwondo-specific aerobic-anaerobic-agility (TAAA) test.**

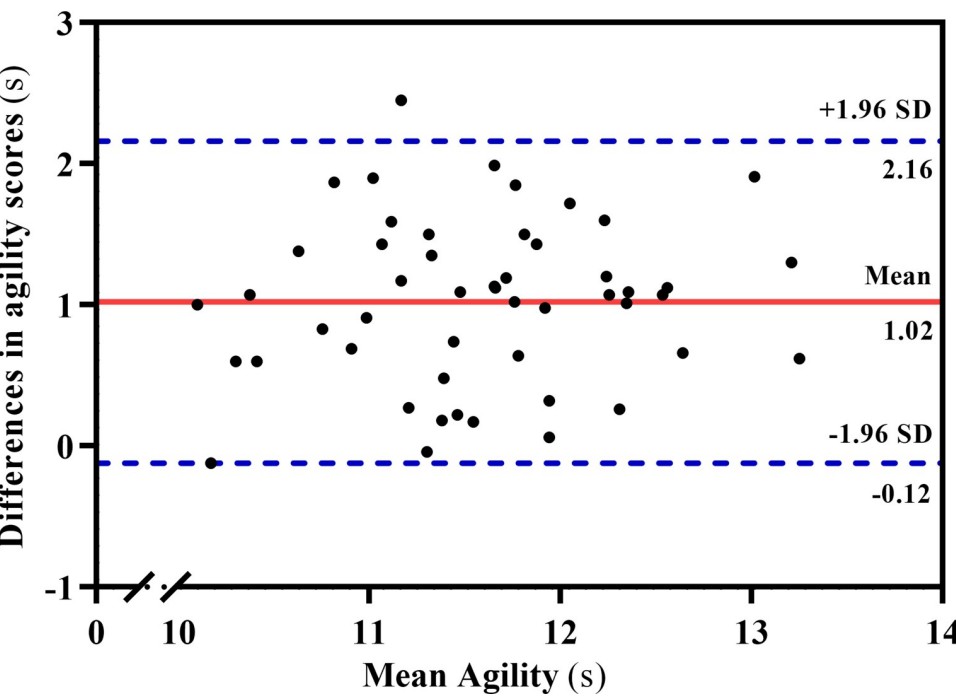

**Fig 4. Bland-Altman plot for comparison between the agility T-Test and the agility function of taekwondo-specific aerobic-anaerobic-agility (TAAA) test.**

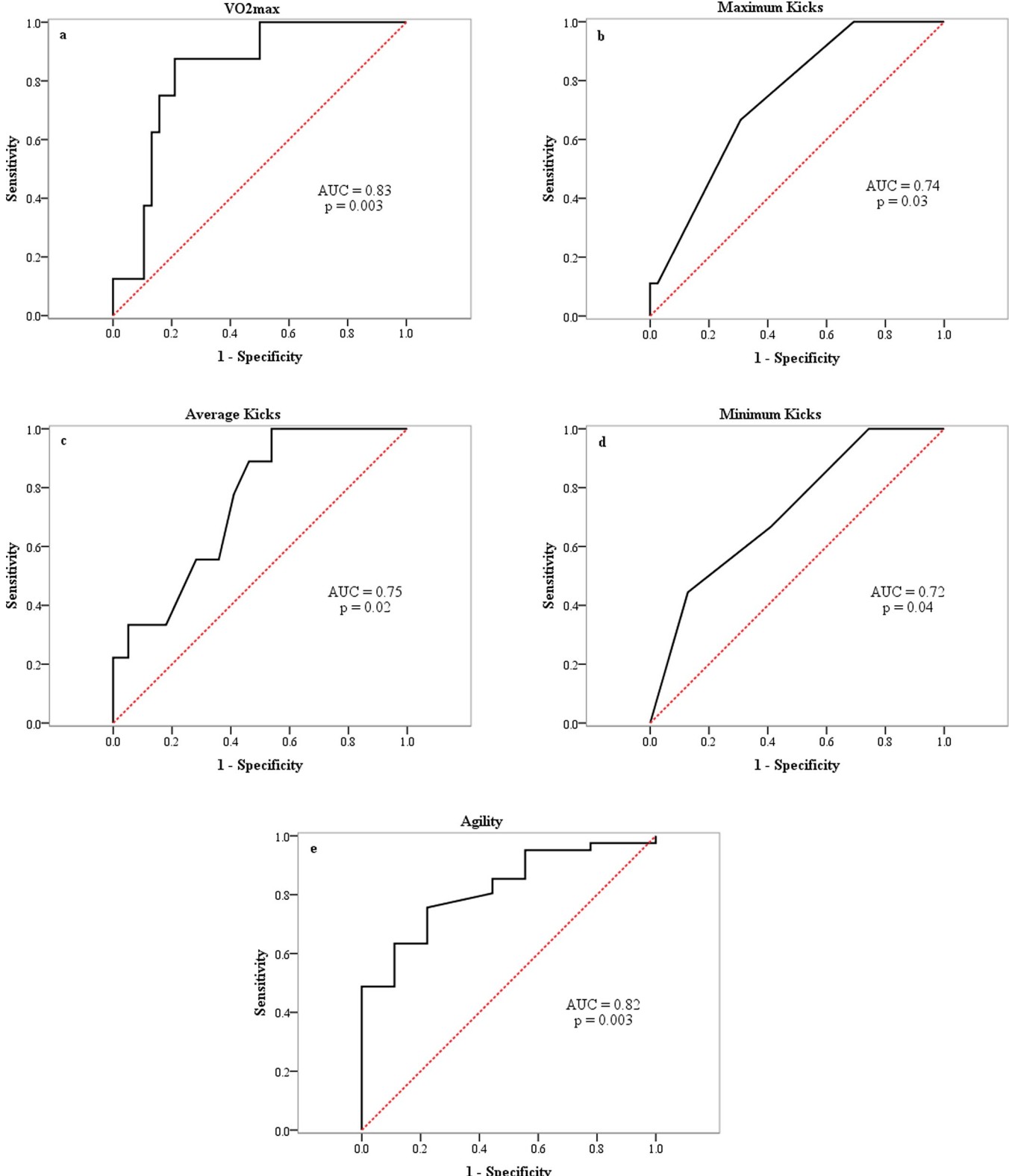

**Fig 5.** Receiver operating characteristics (ROC) curves for a) the predicted values of aerobic power, b) maximum kicks, c) average kicks, d) minimum kicks, and e) agility records between national- and regional-level taekwondo athletes, using the output of the taekwondo-specific aerobic-anaerobic-agility test. AUC: Area under the ROC curve.

sport. To the best of our knowledge, this is the first study to present a specific field test for simultaneously estimating these three important components of fitness. It is widely accepted that competitive taekwondo training places high demands on anaerobic capacity [17,32], but recent studies demonstrate that elite taekwondo athletes have considerable cardio-respiratory fitness [2,3,15,17]. In this study, we have provided an equation that could estimate athletes $VO_{2max}$ using easily measurable variables including BMI, and the difference between HR after and one min after the TAAA test (Eq 2). Although there was an error of about 4.5 mL.kg$^{-1}$.min$^{-1}$, the obtained model explains 70% of the overall variability with the GXT on the treadmill (often regarded as the 'gold standard' test). This value is similar to the model proposed by Rocha et al. [15] (74.3%), but it is noticeably higher than that suggested by Sant'Ana et al. [19] (~39%) derived using progressive taekwondo-specific tests. This indicates a relatively high prediction ability of the variables used to estimate relative $VO_{2max}$ (i.e. BMI and $HR_{diff}$) in a generalized way. The test might therefore offer a convenient, cost effective and time efficient way to broadly estimate the cardio-respiratory fitness of many taekwondo athletes in the field. However, those requiring greater precision of evaluation might favor the use of direct laboratory measurements. Nonetheless, caution is needed and future studies with a larger number of variables to assume that $VO_{2max}$ can actually be predicted with smaller error.

The 20 m multistage shuttle-run test is one of the most common non-specific field-based methods to estimate $VO_{2max}$ in taekwondo athletes [15,19,21,33]. Although this method has been shown to be valid in junior taekwondo athletes [21], the shuttle-run test may underestimate the actual values of $VO_{2max}$ (about 16%) in elite taekwondo athletes [33]. On the other hand, all taekwondo-specific methods proposed in previous studies consist of repeated performance of stationary roundhouse kicks, so that athletes begin with 6 to 10 kicks and then progressively increase 3 to 4 kicks on each new stage until volitional exhaustion [15,17,19]. These tests attempt to broadly reproduce the mode of work performed by athletes during a match, but their main limitations are the lack of common displacements along the competition area and relatively long-duration protocols (i.e. approximately 15 minutes). In addition, another limiting factor of the previous methods compared to TAAA test for estimating $VO_{2max}$ is the use of relative small sample sizes and its homogeneity (only elite black-belt athletes), which in turn may increase the SEE and decrease the generalizability of the results, respectively [19]. Therefore, in our study, the TAAA test experimental design was to eliminate these limitations as much as possible. To this end, the TAAA test included a larger sample than previous studies (46 athletes vs. 14–22 athletes) [15,17,19], common movements performed in competition and training, and shorter time frame (2 min vs. average time>8 min) [15,17], whilst providing estimates of aerobic power.

In the present research, there was a very large and large relationship between relative mean and peak power obtained from WAnT (as a reference test) and average and maximum kicks performed during the TAAA test, respectively, which provided Eq 3 and classificatory Table 3. Furthermore, in order to evaluate the percent decline in kicking performance during the TAAA test, the KFI calculation method was expressed as Eq 1. Considering that taekwondo scoring and defensive actions induce a high demand upon anaerobic metabolic pathways (especially for the phosphagens system), athletes need high anaerobic power abilities to effectively supply the energetic requirements for these actions [3,6,7]. Moreover, peak and mean power play fundamental roles in this regard to produce and sustain high power output via both adenosine triphosphate/creatine phosphate (ATP/PCr) and anaerobic glycolytic metabolic pathways [3]. As such, the present test for determining the anaerobic power parameters could be important across the different athletic levels. Likewise, the equation and classificatory table provided here can help coaches to monitor their athletes' progress and classify their performance for specific goals.

The 30-s WAnT constitutes the most common non-specific method used to assess anaerobic power and capacity of taekwondo athletes, and is widely accepted as a valid anaerobic test [3,16,27]. However, this test lacks mechanical specificity to the actions of taekwondo. Due to a clear need for the development of more specialized anaerobic fitness tests that better represent both the mechanical actions and the anaerobic requirements of the sport, a number of more specific tests have been proposed. These tests have been developed in consideration of the motor skills performed in taekwondo, integrating the execution of stationary roundhouse kicks using a variety of approaches [13,14,16,27,34]. For example, the protocol suggested by two studies [13,16] was designed based on the WAnT and consists of performing the *bandal chagi* technique over 30 s at maximum speed and power against a punching bag, using both legs alternately. In another study, Tayech et al. [27] provided an intermittent kick test that is similar to the running-based anaerobic sprint test (RAST), except that sprints during the RAST are replaced by repeated high-intensity roundhouse kicks (i.e. six 5-s sets, interspersed with 10 s active recovery). Like the RAST protocol, the TAAA test has an intermittent nature with six 20-s of sprints and kicks, interspersed with 10 s rest intervals. This structure was chosen to mimic the real match time that is 2-min rounds with a 1-min rest period between them [28]. Thus, in contrast to the previous intermittent specific tests with stationary nature, the TAAA test may provide a better simulation of taekwondo official matches, as it includes turns and movement across set distances.

Another aspect of the present study is related to agility performance of TAAA test. Given that this test involves specific rotations and change in direction of movement that would be necessary during training and matches, we hypothesized that there may be an association between the time needed to complete six correct kicks in the TAAA test and the records of agility T-Test. The results revealed that there was a very large linear relationship between the records (Pearson's $R = 0.73$, shared variance = 53%). We therefore have presented a classificatory table (Table 3) to help coaches for better monitoring their athletes' agility. A limited number of researchers have studied the agility characteristics of taekwondo athletes using field-based testing methods such as side step tests [12], 50-m (10×5 m) shuttle run sprint tests [35], and agility T-Test [11]. The findings reported by these studies emphasize that agility is an important and effective factor for a better performance in taekwondo. Due to the lack of the specificity of agility assessment and limited ability to discriminate between standards of taekwondo athletes, Chaabene et al. [11] established the validity and reliability of a test of planned agility. This test, like the TAAA test, has a significant large correlation with the T-Test, which is considered by some as a 'reference' planned agility test ($R = 0.71$, shared variance = 50%). Although the mentioned study was the first research that addressed the validity of a taekwondo-specific agility test, the inclusion of athletes only at elite and top-elite level is a limitation that may decrease the generalizability of the proposed test.

Other important aspects which were lacking in previous taekwondo-specific tests validation processes are sensitivity and specificity of the estimations. In the present study, we used the ROC curve method to study the responsiveness of TAAA test in distinguishing between national- and regional-level athletes. The ROC curve that is gaining popularity can be used to determine the responsiveness of performance testing [31]. Based on this method, a performance test would be responsive if its area under the ROC curve was > 0.7 [11,31]. Therefore, this statistical tool reveals that the TAAA test is accurately able to differentiate between national- and regional-level taekwondo athletes (AUC > 0.7; $p < 0.05$).

The current study has some significant merits. On the one hand, a relative big sample size of both black- and red-belt athletes within a big age range was recruited from different martial arts clubs and included to establish the equations for estimating $VO_{2max}$ using the easy-measurable variables obtained from the TAAA test. On the other hand, the anaerobic variables and

agility records of this test had a significant correlation with the reference tests, which the classificatory tables were provided to classify athletes' performance.

Despite the relevance of the present findings, our research has some limitations. Not measuring the impacts of kicking (in order to use simple, available, and inexpensive equipment) and blood lactate, and lack of standardization of the type of displacement (i.e. sprint forwards to the points A and B) during the TAAA test were the main limitations of this study that raise interesting possibilities for future research. With respect to the later, replacing sprint forwards with more taekwondo-specific displacements could be addressed in further studies. Although some recommendations on how to maintain a balance and routine diet over 24 h prior each testing session were provided for the athletes, diet control was not carried out separately. Additionally, we only included adult male athletes. Hence, more investigations might be necessary to develop the TAAA test for estimating the physical fitness components in taekwondo athletes at different classes of age (e.g. adolescents) and in female athletes. It should also be taken into consideration that the reliability of the TAAA method needs to be confirmed before the widespread use of this test. More research therefore is necessary for developing this multidimensional test.

## Conclusions

Several recent studies have developed individual taekwondo-specific tests to estimate or evaluate aerobic and anaerobic fitness, and agility in taekwondo. For the first time, we proposed a new taekwondo-specific field test that is capable of estimating three important fitness components in taekwondo; aerobic power, anaerobic fitness, and agility. The data obtained in the current study revealed that the TAAA test can be considered a valid tool to estimate $VO_{2max}$ and mean power, and to classify anaerobic performance and agility records in taekwondo athletes according to direct comparisons with the reference tests. Importantly, the equations and classificatory tables presented here are useful for coaches and strength and conditioning professionals to classify taekwondo-specific aerobic power, agility and anaerobic fitness. Additionally, the ROC curve method showed that the TAAA test is a simple and convenient tool to accurately distinguish national-level athletes from regional-level.

## Supporting information

**S1 File. PLOS' questionnaire on inclusivity in global research.**
(DOCX)

**S2 File. Excel datasets (xls) for $VO_{2max}$ analysis.** Eq 2 and Fig 5A are released from these datasets.
(XLSX)

**S3 File. Excel datasets (xls) for anaerobic power analysis.** Eq 3, and Fig 5B–5D are released from these datasets.
(XLSX)

**S4 File. Excel datasets (xls) for agility analysis.** Figs 3, 4 and 5E are released from these datasets.
(XLSX)

## Acknowledgments

The authors would like to thank the officials of the Taekwondo Federation of Iran and all athletes and coaches who volunteered to participate in this study. We are grateful to Dr. Majid

Nayeri for his help with the coordination of the athletes and coaches, and to Dr. Samira Gholamian for the technical assistance. We also wish to thank Mr. Alireza Amini and Dr. Majid Taati for their invaluable contribution to the study.

## Author Contributions

**Conceptualization:** Behzad Taati, Hamid Arazi.

**Data curation:** Behzad Taati.

**Formal analysis:** Behzad Taati, Craig A. Bridge, Emerson Franchini.

**Investigation:** Behzad Taati, Hamid Arazi, Craig A. Bridge, Emerson Franchini.

**Methodology:** Behzad Taati, Hamid Arazi, Emerson Franchini.

**Project administration:** Behzad Taati.

**Resources:** Hamid Arazi, Craig A. Bridge.

**Software:** Behzad Taati.

**Supervision:** Hamid Arazi.

**Validation:** Behzad Taati, Hamid Arazi, Emerson Franchini.

**Visualization:** Behzad Taati.

**Writing – original draft:** Behzad Taati, Hamid Arazi.

**Writing – review & editing:** Hamid Arazi, Craig A. Bridge, Emerson Franchini.

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
