## [Decision Letter · Decision Letter 0]

26 Dec 2021

PONE-D-21-34611A new taekwondo-specific field test for estimating aerobic power, anaerobic fitness, and agility performancePLOS ONE

Dear Dr. Arazi,

Thank you for submitting your manuscript to PLOS ONE. After careful consideration, we feel that it has merit but does not fully meet PLOS ONE’s publication criteria as it currently stands. Therefore, we invite you to submit a revised version of the manuscript that addresses the points raised during the review process.

We look forward to receiving your revised manuscript.

Kind regards,

Daniel Boullosa

Academic Editor

PLOS ONE

Journal Requirements:

Reviewers' comments:

Reviewer's Responses to Questions

**Comments to the Author**

1. Is the manuscript technically sound, and do the data support the conclusions?

Reviewer #1: Yes

Reviewer #2: Yes

2. Has the statistical analysis been performed appropriately and rigorously? 

Reviewer #1: Yes

Reviewer #2: Yes

3. Have the authors made all data underlying the findings in their manuscript fully available?

Reviewer #1: Yes

Reviewer #2: Yes

4. Is the manuscript presented in an intelligible fashion and written in standard English?

Reviewer #1: Yes

Reviewer #2: Yes

5. Review Comments to the Author

Reviewer #1: Reviewer Comments

A new taekwondo-specific field test for estimation aerobic power, anaerobic fitness, and agility performance

This study has the originality of creating a Taekwondo-specific fitness test.

[Introduction]

Ln81: “...evaluate aerobic and aerobic power...”

# Are both 'aerobic' correct?

[Methods]

Ln115: “different weight categories”

# You presented no detailed explanation of the “different weight categories.” Why did you use this expression?

Ln124: “being aged 18 to 35 years”

# In this study, the age range of study subjects was wide. I recommend that you describe the age distribution of the study subjects. This is because there is a significant correlation between fitness and age.

Ln219: “bandal tchagui”

# “bandal tchagui” → “bandal chagi”

[Results]

# In this study, you tried to predict aerobic power, anaerobic fitness, and agility using the TAAA test. So you developed the VO2max prediction equation. However, you did not provide a predictive equation for anaerobic fitness or agility performance. I think that your presentation of both prediction equations mentioned above makes the results of this study clear.

[Discussion]

Ln405-408: “Likewise, the classificatory table provided here can help coaches to monitor their athletes’ progress and classify their performance for specific goals. The 30-sec WAnT constitutes the most common non-specific method used to assess anaerobic power and capacity of taekwondo athletes, and is widely accepted as a valid anaerobic test.”

# The classificatory table you presented in this study is helpful, and it would also be beneficial to give a regression model to predict the WAnT test result.

Ln415: “bandal tchagui”

# “bandal tchagui” → “bandal chagi”

# In general, when developing a new test method, developers divide the collected data by 7:3 or 8:2 and then create a test method using 70-80% of the data. And the feasibility of the developed method is checked using the remaining 20-30% of data. However, this study did not use this validation method. Would you please explain why?

Reviewer #2: General comments:

I would like to congratulate the authors for their work. I enjoy the reading. The proposal is very interesting and, in my opinion, has immediate practical applicability. Specifically, the proposal of specific and time-efficient tests that may encourage trainers and athletes to implement evaluations in their daily practice is of great interest. That said, I have just minor comments, as follows:

ABSTRACT

I have no comments

INTRODUCTION

The introduction is well written and successfully drives the reader to the objective. I have no specific comment.

METHODS

Participants

Please, include being more specific about your sample characteristics. How many athletes per weight category?

“some of them were national competitors”. Please, be more specific.

The amount of food ingested was monitored and individualized along with the study. For example, the amount of black tea (Caffeine) was standardized considering athletes' weight and dosage ingested in each attempt? Could it have some influence on the results? What was the interval between the breakfast and testing? It was standardized for all tests?

Statistical analysis

Please provide information about correlations coefficients thresholds and the corresponding reference.

RESULTS

The raw data suggest that only 8 subjects were national-level athletes. Could it affect the results about level distinguishing?

DISCUSSION

The authors considered the “lack of standardization of the type of displacement during the TAAA” as a limitation. Please provide suggestions about how it could be addressed in future studies.

CONCLUSION

In my opinion, the two last sentences of conclusion should be placed with the limitations.

6. PLOS authors have the option to publish the peer review history of their article (what does this mean?). If published, this will include your full peer review and any attached files.

Reviewer #1: No

Reviewer #2: **Yes: **Victor Silveira Coswig

---

## [Author Response · Author response to Decision Letter 0]

11 Jan 2022

Dear Editor and Reviewers

PLOS ONE

We are grateful to you for your time and constructive comments on our manuscript. We have implemented your comments and suggestions that have substantially helped us in improving our manuscript, thank you for that. We would now wish to submit a revised version of the manuscript for further consideration in the journal. Changes in the text are yellow and green coloured in the revised version. We hope that our revised manuscript has improved in form and content and is now ready for publication in PLOS ONE. 

Yours sincerely,

Response to the Reviewer 1

A new taekwondo-specific field test for estimation aerobic power, anaerobic fitness, and agility performance.

This study has the originality of creating a Taekwondo-specific fitness test.

[Introduction]

Ln81: “...evaluate aerobic and aerobic power...”

# Are both 'aerobic' correct?.

Response: This was a typo and we therefore revised it, i.e.: 

“to evaluate aerobic and anaerobic power”

[Methods]

Ln115: “different weight categories”

# You presented no detailed explanation of the “different weight categories.” Why did you use this expression?

Ln124: “being aged 18 to 35 years”

# In this study, the age range of study subjects was wide. I recommend that you describe the age distribution of the study subjects. This is because there is a significant correlation between fitness and age.

Response: We provided an additional table to present the distribution of athletes in different weight and age categories (Table 2). This table is based on the competition rules provided by World Taekwondo Federation [9]. 

1. World Taekwondo Federation. Competition Rules and Interpretation 2020 [Available from: http://www.worldtaekwondo.org/viewer_pdf/external/pdfjs-2.1.266-dist/web/viewer.html?file=http://www.worldtaekwondo.org/att_file/documents/WT%20Competition%20Rules_Interpretation%20(October%201,%202020).pdf.

Ln219: “bandal tchagui”

# “bandal tchagui” → “bandal chagi”

Response: “bandal tchagui” was replaced with “bandal chagi” throughout the main text. 

[Results]

# In this study, you tried to predict aerobic power, anaerobic fitness, and agility using the TAAA test. So you developed the VO2max prediction equation. However, you did not provide a predictive equation for anaerobic fitness or agility performance. I think that your presentation of both prediction equations mentioned above makes the results of this study clear.

Response: We have proposed a predictive equation for VO2max based on the regression model between the TAAA test and a standard protocol on treadmill using a calibrated metabolic analyzer. We know that a validated predictive equation like this can be very useful during training and preparation period. However, it is questionable for us what are the practical benefits of predicting Wingate test power or T-Test record based on the TAAA test. Indeed, we think the interest here is to predict taekwondo-related performances based on a valid specific test (i.e. the TAAA test) and there is no functional necessity to provide predictive equation here. We therefore decided to develop a classificatory table to provide a clear scale for comparing the athletes based on the TAAA test outputs. 

[Discussion]

Ln405-408: “Likewise, the classificatory table provided here can help coaches to monitor their athletes’ progress and classify their performance for specific goals. The 30-sec WAnT constitutes the most common non-specific method used to assess anaerobic power and capacity of taekwondo athletes, and is widely accepted as a valid anaerobic test.”

# The classificatory table you presented in this study is helpful, and it would also be beneficial to give a regression model to predict the WAnT test result.

Response: This table is a useful scale for monitoring taekwondo athletes’ performances based on the number of kicks and agility records. In our opinion, this classificatory table is more applicable than the regression model. As discussed above, we think there is no functional necessity to predict the results of Wingate test and agility T-Test using the TAAA test outputs (i.e. the number of kicks and the agility records). However, significant correlation between the tests was needed to validate the TAAA test, as performed in the present study. 

Ln415: “bandal tchagui”

# “bandal tchagui” → “bandal chagi”

Response: “bandal tchagui” was replaced with “bandal chagi” throughout the main text. 

# In general, when developing a new test method, developers divide the collected data by 7:3 or 8:2 and then create a test method using 70-80% of the data. And the feasibility of the developed method is checked using the remaining 20-30% of data. However, this study did not use this validation method. Would you please explain why?

Response: The validation method applied in the present study was based on the previous works in this area [1-8]. Because no study has used the method you explained here, we did not perform it. However, if you insist on using this method, we should change all data analysis and probably all the results. 

1. Araujo MP, Nóbrega AC, Espinosa G, Hausen MR, Castro RR, Soares PP, et al. Proposal of a new specific cardiopulmonary exercise test for taekwondo athletes. J Strength Cond Res. 2017;31(6):1525-35.

2. Chaabene H, Negra Y, Capranica L, Bouguezzi R, Hachana Y, Rouahi MA, et al. Validity and reliability of a new test of planned agility in elite taekwondo athletes. J Strength Cond Res. 2018;32(9):2542-7.

3. Chaabène H, Hachana Y, Franchini E, Mkaouer B, Montassar M, Chamari K. Reliability and construct validity of the karate-specific aerobic test. J Strength Cond Res. 2012;26(12):3454-60.

4. Rocha F, Louro H, Matias R, Costa A. Anaerobic fitness assessment in taekwondo athletes: A new perspective. Motricidade. 2016;12(2):127-39.

5. Rocha FP, Louro H, Matias R, Brito J, Costa AM. Determination of Aerobic Power Through a Specific Test for Taekwondo-A Predictive Equation Model. J Hum Kinet. 2016;53(1):117-26.

6. Sant’Ana J, Franchini E, Sakugawa RL, Diefenthaeler F. Estimation equation of maximum oxygen uptake in taekwondo specific test. Sport Sci Health. 2018;14(3):699-703.

7. Tayech A, Mejri MA, Chaabene H, Chaouachi M, Behm DG, Chaouachi A. Test-retest reliability and criterion validity of a new Taekwondo Anaerobic Intermittent Kick Test. The Journal of sports medicine and physical fitness. 2018;59(2):230-7.

8. Tayech A, Mejri MA, Chaouachi M, Chaabene H, Hambli M, Brughelli M, et al. Taekwondo anaerobic intermittent kick test: Discriminant validity and an update with the gold-standard wingate test. J Hum Kinet. 2020;71(1):229-42.

Kind regards,

Response to the Reviewer 2

Dear Reviewer

We are grateful to you for your time and constructive comments on our manuscript. We have implemented your comments and suggestions that have substantially helped us in improving our manuscript, thank you for that. We would now wish to submit a revised version of the manuscript for further consideration in the journal. Changes in the text are green coloured in the revised version. We hope that our revised manuscript has improved in form and content and is now ready for publication in PLOS ONE. 

Yours sincerely,

I would like to congratulate the authors for their work. I enjoy the reading. The proposal is very interesting and, in my opinion, has immediate practical applicability. Specifically, the proposal of specific and time-efficient tests that may encourage trainers and athletes to implement evaluations in their daily practice is of great interest. That said, I have just minor comments, as follows:

ABSTRACT

I have no comments.

INTRODUCTION

The introduction is well written and successfully drives the reader to the objective. I have no specific comment.

METHODS

Participants

Please, include being more specific about your sample characteristics. How many athletes per weight category?

Response: We provided an additional table to present the distribution of athletes in different weight and age categories (Table 2). This table is based on the competition rules provided by World Taekwondo Federation [1]. 

1. World Taekwondo Federation. Competition Rules and Interpretation 2020 [Available from: http://www.worldtaekwondo.org/viewer_pdf/external/pdfjs-2.1.266-dist/web/viewer.html?file=http://www.worldtaekwondo.org/att_file/documents/WT%20Competition%20Rules_Interpretation%20(October%201,%202020).pdf.

“some of them were national competitors”. Please, be more specific.

Response: More details were added to this section, i.e.:

“Nine athletes were national competitors (i.e. a member of the national taekwondo team).”

The amount of food ingested was monitored and individualized along with the study. For example, the amount of black tea (Caffeine) was standardized considering athletes' weight and dosage ingested in each attempt? Could it have some influence on the results? What was the interval between the breakfast and testing? It was standardized for all tests?

Response: We added this point to the limitations of our study, i.e.: “Although some recommendations on how to maintain a balance and routine diet over 24 h prior each testing session were provided for the athletes, diet control was not carried out separately”.

 It should also be noted that as we explained before in the “Graded exercise test” and “Wingate anaerobic test” sections, in order to standardize the later meal, all athletes were instructed to eat their breakfast until 6:30 am (about 4 hours prior to commencing testing), which included 2-3 units of bread (�50-75 g), 1 unit of white cheese (�10 g), and 1 cup of black tea containing 1 teaspoon of white sugar. They were also refrained from consuming any other meals after the breakfast until the end of the test.

Statistical analysis

Please provide information about correlations coefficients thresholds and the corresponding reference.

Response: More details and the corresponding reference were added to this section according to your comment, i.e.:

“The magnitude of R was considered as trivial (< 0.1), small (0.1 < R < 0.3), moderate (0.3 < R < 0.5), large (0.5 < R < 0.7), very large (0.7 < R < 0.9), nearly perfect (R = 0.9), and perfect (R = 1) [2].”

2. Hopkins WG. A scale of magnitudes for effect statistics 2009 [411]. Available from: http://sportsci.org/resource/stats/effectmag.html.

RESULTS

The raw data suggest that only 8 subjects were national-level athletes. Could it affect the results about level distinguishing?

Response: Overall, nine national-level athletes participated in this study. As mentioned in the “External Responsiveness of TAAA test” section, nine athletes completed agility and power assessments, and eight athletes completed aerobic assessments. We included the athletes from different levels to increase the generalizability of the results. Therefore, nine (18%) and eight (16%) national-level athletes can be a relative good sample of elite taekwondo athletes’ population based on the normal distribution theory.

DISCUSSION

The authors considered the “lack of standardization of the type of displacement during the TAAA” as a limitation. Please provide suggestions about how it could be addressed in future studies.

Response: In our opinion, sprint forwards to the points A and B may be non-specific in terms of type of displacement during taekwondo competitions. We therefore considered it as a limitation and the relevant suggestion was provide for future studies, i.e.:

“With respect to the later, replacing sprint forwards with more taekwondo-specific displacements could be addressed in further studies”

CONCLUSION

In my opinion, the two last sentences of conclusion should be placed with the limitations.

Response: These two sentences were placed with the limitations based on your accurate comment. 

Kind regards,

Hamid Arazi

Corresponding author of the paper

---

## [Decision Letter · Decision Letter 1]

9 Feb 2022

PONE-D-21-34611R1A new taekwondo-specific field test for estimating aerobic power, anaerobic fitness, and agility performancePLOS ONE

Dear Dr. Arazi,

Thank you for submitting your manuscript to PLOS ONE. After careful consideration, we feel that it has merit but does not fully meet PLOS ONE’s publication criteria as it currently stands. Therefore, we invite you to submit a revised version of the manuscript that addresses the points raised during the review process.

 Please, address the reviewer comments the simpler manner you can.

We look forward to receiving your revised manuscript.

Kind regards,

Daniel Boullosa

Academic Editor

PLOS ONE

Journal Requirements:

Reviewers' comments:

Reviewer's Responses to Questions

**Comments to the Author**

1. If the authors have adequately addressed your comments raised in a previous round of review and you feel that this manuscript is now acceptable for publication, you may indicate that here to bypass the “Comments to the Author” section, enter your conflict of interest statement in the “Confidential to Editor” section, and submit your "Accept" recommendation.

Reviewer #1: (No Response)

Reviewer #2: All comments have been addressed

2. Is the manuscript technically sound, and do the data support the conclusions?

Reviewer #1: Partly

Reviewer #2: Yes

3. Has the statistical analysis been performed appropriately and rigorously? 

Reviewer #1: No

Reviewer #2: Yes

4. Have the authors made all data underlying the findings in their manuscript fully available?

Reviewer #1: Yes

Reviewer #2: Yes

5. Is the manuscript presented in an intelligible fashion and written in standard English?

Reviewer #1: Yes

Reviewer #2: Yes

6. Review Comments to the Author

Reviewer #1: Authors must distinguish between 'practical methods' and 'correct methods.' Are the classificatory tables developed by the authors more accurate than Wingate test results? Or could the classificatory table created by the authors replace the Wingate test results? The result of an estimate always has an error. That is why developers should try to reduce this error.

I expect this method created by the authors to be practically used in the Taekwondo field. So, to further reduce the error of this classification table, I proposed the estimation formulas. However, I felt that the authors disagreed with my suggestion that the findings of this study should be made more valid. Therefore, I can't entirely agree with the publication of this paper because I expect that the error of the results of this study will be significant.

Reviewer #2: Thank you for considering my contribution. All comments have been adressed and I have no further suggestions. Congratulations.

7. PLOS authors have the option to publish the peer review history of their article (what does this mean?). If published, this will include your full peer review and any attached files.

Reviewer #1: No

Reviewer #2: **Yes: **Victor Silveira Coswig

---

## [Author Response · Author response to Decision Letter 1]

10 Feb 2022

Dear Reviewer

We are so thankful for your accurate comments in order to reduce existing errors. We have applied your comment in the article, which can be found in yellow color.

Authors must distinguish between 'practical methods' and 'correct methods.' Are the classificatory tables developed by the authors more accurate than Wingate test results? Or could the classificatory table created by the authors replace the Wingate test results? The result of an estimate always has an error. That is why developers should try to reduce this error.

I expect this method created by the authors to be practically used in the Taekwondo field. So, to further reduce the error of this classification table, I proposed the estimation formulas. However, I felt that the authors disagreed with my suggestion that the findings of this study should be made more valid. Therefore, I can't entirely agree with the publication of this paper because I expect that the error of the results of this study will be significant.

Response: The regression model was applied to provide the equation that accurately describes the relationships between peak and mean power in the Wingate test and the results of the TAAA test, i.e.: 

“The regression models were therefore used to describe the relationship between the correlated variables. The KS test was applied to detect the normality assumption of residuals for maximum (KS = 0.23, p < 0.001) and average (KS = 0.09, p = 0.2) values. Because this assumption was not met for peak power, the regression model for this variable was not accepted. However, a linear regression model was developed to estimate average power values according to the relationship between average power output obtained from WAnT and average kicks in the TAAA test (t = 8.81; p < 0.001), in which the equation that best described this relationship was as follows:

Equation (3)

Average Power (W.kg-1) = 0.648 (average kicks)

The abovementioned equation could explain 63% of the overall variability between the variables with a SEE of 0.53 W.kg-1. The mean and SD of average power for the TAAA test and WAnT were 7.3 � 0.85 and 7.3 � 0.68 W.kg-1, respectively. The mean CV was 11.7% and 9.3% for the TAAA test and WAnT, respectively. We then performed the ANOVA Levene´s F test to confirm there was no difference in the CV between the tests (F = 2.34; p = 0.12).”

Kind regards,

---

## [Decision Letter · Decision Letter 2]

22 Feb 2022

A new taekwondo-specific field test for estimating aerobic power, anaerobic fitness, and agility performance

PONE-D-21-34611R2

Dear Dr. Arazi,

We’re pleased to inform you that your manuscript has been judged scientifically suitable for publication and will be formally accepted for publication once it meets all outstanding technical requirements.

Kind regards,

Daniel Boullosa

Academic Editor

PLOS ONE

Additional Editor Comments (optional):

Reviewers' comments:

Reviewer's Responses to Questions

**Comments to the Author**

1. If the authors have adequately addressed your comments raised in a previous round of review and you feel that this manuscript is now acceptable for publication, you may indicate that here to bypass the “Comments to the Author” section, enter your conflict of interest statement in the “Confidential to Editor” section, and submit your "Accept" recommendation.

Reviewer #1: All comments have been addressed

Reviewer #2: All comments have been addressed

2. Is the manuscript technically sound, and do the data support the conclusions?

Reviewer #1: Yes

Reviewer #2: Yes

3. Has the statistical analysis been performed appropriately and rigorously? 

Reviewer #1: Yes

Reviewer #2: Yes

4. Have the authors made all data underlying the findings in their manuscript fully available?

Reviewer #1: Yes

Reviewer #2: Yes

5. Is the manuscript presented in an intelligible fashion and written in standard English?

Reviewer #1: Yes

Reviewer #2: Yes

6. Review Comments to the Author

Reviewer #1: There is no explanation for the predictive model of anaerobic power in the statistical analysis section. Please add it.

[VO2max (mL.kg-1.min-1) = 56.316 + 0.742 (HRdiff) – 0.924 (BMI)] vs. [Average Power (W.kg-1) = 0.648 (average kicks)]

As a result of comparing the regression equations presented by you, there is no y-intercept of the prediction equation of Average Power. You must check that this equation is expressed correctly.

Overall, I am very impressed that you have developed a field test specific to Taekwondo. So I hope that Taekwondo athletes will put the test you created to good use.

Reviewer #2: I have no further comments. Thank you for the opportunity to contribute.

7. PLOS authors have the option to publish the peer review history of their article (what does this mean?). If published, this will include your full peer review and any attached files.

Reviewer #1: **Yes: **Sang-Seok Nam

Reviewer #2: **Yes: **Victor Silveira Coswig

---

## [Editor Report · Acceptance letter]

7 Mar 2022

PONE-D-21-34611R2 

A new taekwondo-specific field test for estimating aerobic power, anaerobic fitness, and agility performance 

Dear Dr. Arazi:

I'm pleased to inform you that your manuscript has been deemed suitable for publication in PLOS ONE. Congratulations! Your manuscript is now with our production department. 

Kind regards, 

on behalf of

Dr. Daniel Boullosa 

Academic Editor

PLOS ONE